

# Privacy preserved and decentralized thermal comfort prediction model for smart buildings using federated learning

Sidra Abbas[1], Shtwai Alsubai[2], Gabriel Avelino Sampedro[3,4], Mideth Abisado[5], Ahmad Almadhor[6] and Tai-hoon Kim[7]

[1] Department of Computer Science, COMSATS University, Islamabad, Sahiwal, Pakistan
[2] College of Computer Engineering and Sciences, Prince Sattam bin Abdulaziz University, AlKharj, Saudi Arabia
[3] Faculty of Information and Communication Studies, University of the Philippines Open University, Los Baños, Philippines
[4] Center for Computational Imaging and Visual Innovations, De La Salle University, Malate, Philippines
[5] College of Computing and Information Technologies, National University, Manila, Philippines
[6] Department of Computer Engineering and Networks, College of Computer and Information Sciences, Jouf University, Sakaka, Saudi Arabia
[7] School of Electrical and Computer Engineering, Yeosu Campus, Chonnam National University, Yeosu-si, Jeollanam-do, Republic of South Korea

Corresponding authors
Sidra Abbas, sidraabbas@ieee.org
Tai-hoon Kim,
taihoonn@chonnam.ac.kr,
taihoonn@empas.com

## ABSTRACT

Thermal comfort is a crucial element of smart buildings that assists in improving, analyzing, and realizing intelligent structures. Energy consumption forecasts for such smart buildings are crucial owing to the intricate decision-making processes surrounding resource efficiency. Machine learning (ML) techniques are employed to estimate energy consumption. ML algorithms, however, require a large amount of data to be adequate. There may be privacy violations due to collecting this data. To tackle this problem, this study proposes a federated deep learning (FDL) architecture developed around a deep neural network (DNN) paradigm. The study employs the ASHRAE RP-884 standard dataset for experimentation and analysis, which is available to the general public. The data is normalized using the min-max normalization approach, and the Synthetic Minority Over-sampling Technique (SMOTE) is used to enhance the minority class's interpretation. The DNN model is trained separately on the dataset after obtaining modifications from two clients. Each client assesses the data greatly to reduce the over-fitting impact. The test result demonstrates the efficiency of the proposed FDL by reaching 82.40% accuracy while securing the data.

# INTRODUCTION

The emergence of digital revolutions has made it possible for smart cities to offer amenities like intelligent healthcare systems, secure neighborhoods, smart homes, smart transit, and smart buildings (*Javed et al., 2022*; *Rehman et al., 2022*). In smart buildings, smart homes/rooms are the optimal choices to monitor and assess the health of residents (*Javed et al., 2021*). The smart building includes internal kitchen appliances and subsystems for

illumination, monitoring, heating, ventilation, air conditioning (HVAC), water supply, *etc.* These components are regularly modified to provide the user with energy and comfort. Energy consumption forecasts for such intelligent buildings are important owing to the intricate decision-making processes surrounding resource efficiency. The anticipated energy can benefit intelligent buildings by improving their use of energy. In other terms, the building management system may automatically reduce energy costs by employing energy prediction without affecting user comfort (*Fayyaz, Farhan & Javed, 2022*; *Petidis et al., 2018*; *Sülo et al., 2019*). Energy forecasts can support smart grid energy budgeting. The stability of the grid is affected by peak usage during particular periods. Therefore, smart grid technology must forecast and schedule the demand in real-time to accomplish the electricity production balance (*Almalaq et al., 2019*; *Pérez-Lombard, Ortiz & Pout, 2008*). Using past information, the prediction systems must accurately anticipate the power consumption for a certain period and under various environmental circumstances.

The development of artificial intelligence and data analytics has led to the discussion of numerous machine learning (ML) based building energy consumption forecast algorithms such as for human thermal comfort presented in *Farhan et al. (2015)*, *Luo et al. (2018)*, *Kim, Schiavon & Brager (2018)* and *Auffenberg, Stein & Rogers (2015)*. Numerous methods have been implemented to optimize environmental sustainability and regulate building heat (*Dounis & Caraiscos, 2009*; *Dounis et al., 1996*; *Shepherd & Batty, 2003*; *Calvino et al., 2004*; *Maasoumy, Pinto & Sangiovanni-Vincentelli, 2011*). However, it has been found that for any ML system to perform well, it needs to be trained on large amounts of data.

The privacy of information providers may be jeopardized because the sensitive data needed to train the ML algorithms is needed (*Dasari et al., 2021*; *Lee, Xie & Choi, 2021*). While ML algorithms are typically performed remotely on the cloud, Internet of Things (IoT) devices and applications are typically installed in our homes and places of employment (*Majid et al., 2022*; *Razip et al., 2022*; *Chatterjee, Kar & Mustafa, 2021*). As a result, the data supplied with the cloud services may be used immorally. Data sharing can be problematic even within a company due to concerns about privacy, competitiveness, ethics, *etc*. These privacy concerns worsen when cloud services are used and collaborative data must be handled. These privacy violations depend heavily on the data's commercial worth. When employing cloud-based machine learning tools, private information must be managed safely, discreetly, and morally to avoid serious privacy violations (*Dasari et al., 2021*). In addition to rules, privacy-enhanced ML techniques, such as federated learning techniques, are crucial in mitigating ethical and privacy violations (*Moradzadeh et al., 2021*; *Cheng, Li & Liu, 2022*).

Federated learning (FL) was first used by Google (*Aledhari et al., 2020*) and *Li et al. (2020)* to describe a cooperative strategy for enhancing privacy. It is a distributed machine learning platform that protects privacy, with the model accessing the data rather than vice versa. The fundamental concept behind this framework is to prevent data leaking by creating ML models from datasets stored on the data owners' machines. Such distributed data architectures are synchronizing due to advancements in edge processing.

This study presents an FDL method based on a DNN model. In an FDL structure, the recommended framework trains all clients' local frameworks before transmitting the global

framework attributes. The main model, which ensures the confidentiality of the client's data, is used to start the training process after the new parameter has been aggregated.

## Contribution

The research's primary contribution is presented in the list format below.

- This article proposed an approach for the thermal comfort model that enhances predictions of thermal experience by concentrating on important features using the ASHRAE RP-884 standard dataset. The novelty lies in the privacy preservation of the individual's data and improved performance using federated learning.
- The proposed approach first builds the local model and then delivers the variables of the global model. Subsequently, the global model integrates the revised variables and enhances the process while preserving the privacy of every client's information.
- The proposed method employs the min-max normalization technique to normalize the data and leverages SMOTE to enhance the depiction of the minority class's perception. The global model receives its parameters from the local model, which is trained initially through the FDL paradigm. Next, the global model ensures that each client's data is private by aggregating the new parameters and training the system.
- The experiment demonstrates that the FDL model is based on a deep neural network, reaching 82.40% accuracy while offering client data protection.

## Organization

The following sections comprise the article: The heat balancing and ML and FL methodologies for analyzing the thermal comfort model are discussed in 'Related Work'. 'Federated Deep Learning Framework' explains the proposed work's research methodology using the ASHRAE RP-884 standard dataset and DNN model. The results are explained and discussed in 'Experimental Results and Analysis'. 'Conclusion and Future Scope' contains the study's summary and recommendations for further research.

## RELATED WORK

Energy consumption forecasts for intelligent buildings are essential owing to the intricate decision-making procedure surrounding resource efficiency. In the literature, numerous models account for building energy consumption forecasts (*Ren et al., 2023*; *Han et al., 2023*; *Jaffal, 2023*; *Wang, Hu & Chen, 2023*). The literature is divided into three categories: machine learning techniques, federated learning techniques to predict thermal comfort sensations and heat balance approaches.

*Heat balancing techniques:* Implement a simulation system for building thermal control and test the performance in various configurations to gauge the performance (*Rehman et al., 2022*; *Petidis et al., 2018*). The author presents the building thermal control in *Gao, Li & Wen (2019)* as a cost-minimization issue that considers both the inhabitants' thermal comfort and the energy use of the HVAC system. The experiment's findings demonstrate that the proposed method may increase the precision of thermal comfort predictions, lower HVAC energy usage, and increase thermal comfort for occupants. The occupants'

contentment with the thermal environment is reflected in their thermal comfort *ASHRAE (1992)*.

Thermal comfort models are proposed to forecast the occupants' happiness under specific thermal conditions to evaluate thermal comfort objectively (*Luo et al., 2018*; *Auffenberg, Stein & Rogers, 2015*). Thermal comfort is evaluated subjectively since the occupants' subjective experiences in a specific thermal environment. Many participants will be asked to rate how satisfied they are in a variety of thermal situations, including neutral (0), cool (-1), warm (1), cold (-2), and hot (2). The data can then be fitted using some mathematical or heuristic methods. A variety of models, including Predicted Mean Vote (PMV), Actual Mean Vote (AMV), Predicted Percentage Dissatisfied (PPD), and others, have been developed (*Cheng, Niu & Gao, 2012*) to assess customers' thermal satisfaction under various thermal conditions.

Different transfer learning techniques are proposed in *Pinto et al. (2022)*, *Khalil et al. (2021)* and *Somu et al. (2021)* to decrease power consumption in intelligent structures. When there is insufficient previous training data, *Khalil et al. (2021)* uses a novel transfer learning technique to improve occupancy forecast accuracy. Three office spaces in an educational institution are used as a case study for the proposed strategy and models. The data sets employed in this study are gathered from Newcastle University's Urban Sciences Building (USB). ML algorithms have been compared to the outcomes of the proposed transfer learning technique. The findings demonstrate that the proposed model accurately works.

*Machine learning techniques:* ML techniques consider behavioral, psychological, and physiological variables for thermal comfort (*Fayyaz, Farhan & Javed, 2022*; *Farhan et al., 2015*). The author utilizes deep learning and a time-series-based technique to predict temperature preferences by framing the challenge as a multivariate, multi-class classification problem. Integrate a normalized long short term (LSTM) with L1 regularization. Using a memory network to prevent overfitting and use this model's attentional processes. An experiment is conducted on fourteen issues, and evaluation measurements are utilized. The proposed model performed well with 78% accuracy (*Chennapragada et al., 2022*).

The researchers in *Cakir & Akbulut (2022)* aims to assist facility managers in anticipating thermal sensations under conditions. For prediction, a data-driven methodology is used on the gathered dataset. The proposed methodology is based on a deep neural network algorithm. The Bayesian algorithm is employed to optimize the hyperparameter of the deep neural network method. The proposed methodology, with an accuracy of 78%, performed well compared to traditional approaches.

The author in *Deng & Chen (2021)* proposes a rule reinforcement learning model to explain how people change their clothes and thermostat settings. Modeling of user behavior using a Markov decision process. The MDP includes user behavior and several impact factors in the activity and state space. The user actions are intended to create a friendlier atmosphere. The author presents the total variation in the thermal feeling vote beforehand and following the alterations action as an incentive for completing the task. Q-learning is employed to train the model on the used dataset. Following training, the

model correctly predicted behavior involving thermostat set point adjustments with an R2 of 0.75 to 0.8.

*Federated learning techniques:* Management systems for intelligent buildings must handle energy effectively. Energy forecasting is crucial to accomplishing this. Predicting energy use is helpful for intelligent grid power, consumer spending budgeting, and comfort control in intelligent buildings. Machine learning methods are used to estimate energy usage in general. To be effective, machine learning algorithms need many data, however. The collection of this information from data owners may result in privacy violations. Recent years have seen a paradigm shift in machine learning due to growing worries about data privacy and ownership. Federated learning (FL), an emerging paradigm, is now a cutting-edge design for machine learning applications (*Cheng, Li & Liu, 2022*; *Aledhari et al., 2020*).

To protect the confidentiality of power scheduling of building structures related to a shared energy storage system (SESS), the author in *Lee, Xie & Choi (2021)* presents a distributed deep reinforcement learning (DRL) framework by using the federated reinforcement learning (FRL) method, which comprised of a global server (GS) as well as local building energy management systems (LBEMSs). According to the paradigm, the GS without consumer energy consumption data receives just a randomly chosen portion of the trained NN for power consumption methods from the LBEMS DRL agents. The findings show that the proposed method can plan the charging and discharge of the SESS and efficient energy use of HVAC systems in intelligent buildings.

The researcher presents a novel method for predicting heating load demand in *Moradzadeh et al. (2021)* built on cyber-secure federated deep learning (CSFDL). The proposed CSFDL offers a worldwide supermodel for predicting the requirement for heating load from various local clients without identifying their location and, most significantly, without disclosing their privacy. The study trains and tests a CSFDL global server while considering the ventilation system's thermal load requirements of ten different customers. It was shown that the worldwide supermodel had a 98.00%, 93.00%, and 70.00% correlation coefficient for predicting the need for heating.

A novel use of the federated learning framework is provided in research *Li et al. (2020)*, *Perry & Fallon (2019)*, with an intelligent building energy forecast scenario as the focal point. Deep neural networks are employed with data from the ASHRAE to implement and assess this design. The applicability of the federated learning framework is given by its architectural features. The effectiveness of this forecasting model in comparison to the centralized machine learning technique is also addressed.

Heat balancing methods have some issues that cannot be disregarded. They conducted their study only on one age group, primarily in an office setting. Furthermore, they should have examined how location, age, and weather impacted their findings. Additionally, the proposed model is static and does not evolve in response to environmental changes. In essence, many limits are being addressed by utilizing heat balancing approaches, machine learning techniques, and a range of datasets to test the generalizability of methodologies.

# FEDERATED DEEP LEARNING FRAMEWORK

Deep learning models have significantly increased in popularity over the last decade and are now employed in many fields. Due to expanding data availability and increased computer capacity, more effective deep learning algorithms have given rise to new applications, such as smart buildings, automated driving, intelligent energy systems, *etc.*

Usually, these algorithms cannot function accurately unless a significant quantity of data is provided. Usually, all this data is kept on a central server, which could prove problematic for the network. First, it is simple to access these servers, making them an accessible target for attackers, given the expanding bandwidth availability and the potential for remote database systems. Second, analyzing data collected in enormous quantities on a single server takes longer. Last, devices linked to servers may gather sensitive data that must be kept private according to laws and regulations. It is crucial to use more effective methods of decentralized data processing to address these issues. The FDL, which Google recently unveiled (*Aledhari et al., 2020*; *Li et al., 2020*), is used in this research.

FDL anticipates a centralized global framework to combine DL models with local customers without revealing their location or, more crucially, their confidentiality. This approach seeks to build a global model with a federation of numerous collaborating machines that maintain security over their information in the presence of typically numerous clients with Internet connections. Additionally, this approach requires less computational processing and further works without a central server to receive data from nearby parties. Following each iteration process, a broadcast model is retrieved from the cloud's primary server, after which the local parties instruct their local information and transfer modified values back to the server for the following iteration. On the server side, integration is employed to get the new global model.

In a distributed structure, the FDL approach considers the M-allowed clients and lets m represent the index of each client to calculate the basic loss function analytically as shown in Eqs. (1) and (2):

$$min_\theta l(\theta) = \sum_{m=1}^{M} \frac{n_m}{n} K_m(\theta) \tag{1}$$

where each client's local samples are $n_m$ and the $K_m(\theta)$ can write as:

$$K_m(\theta) = \frac{1}{n_m} \Sigma_{j \varepsilon R_k} L_q(\theta) \tag{2}$$

where dataset indexes are $R_k$ and $n_m$ is the length. The FDL framework consists of one server and two clients, represented in Fig. 1. The procedure consists of four crucial steps. First, basic training is carried out in every local party using the parameters $\theta_k$ received from the remote server. In the second stage, called model aggregating, the server uses each client's parameters $\theta_k^u$ to conduct a reliable aggregate over the clients. The third stage is parameter transmission, in which the server sends updated parameters $\theta_{t+1}$ to every local client so they can retrain by employing their dataset. Finally, each client upgrades

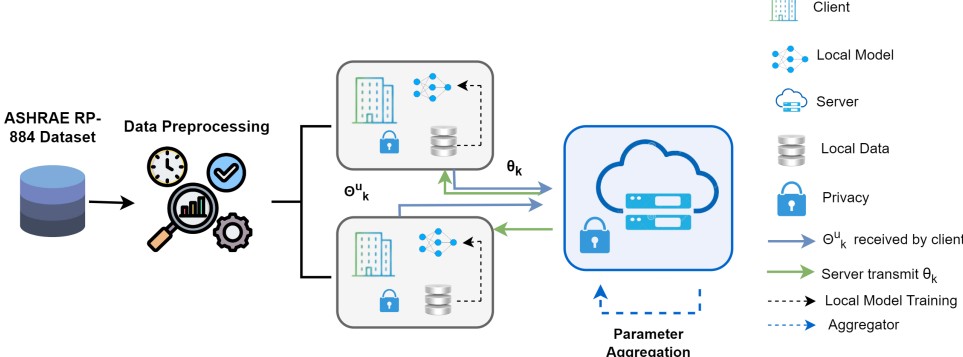

**Figure 1** **Proposed federated deep learning framework overview.**

their distinct models given the combined parameters, and the effectiveness of the updated models is assessed.

The training dataset was separated into two clients, and the dataset $D_s$ was cleansed initially in Algorithm 1. Initialize the model weight ($w_0$) and upgrade the global model. R is the entire round of the local model, and $r_i$ is the model's existing round. The total number of clients is C, and the present client is $c_i$. For every client, change the local model based on the iteration/round that is currently underway. Once the value of the client's data and present client round are added together, determine the weight of the present repetition. For every client, the degradation of the local model is computed using the model variables, epoch value, activation function, and batch size. Compute $F_i(w)$, the loss function, and update the local model. This process continues until the required precision is reached or the loss function is continuously minimized.

---

**Algorithm 1** Algorithm for Federated Deep Learning Model

---

1: $D_s$ ASHRAE RP-884 dataset

2: $D_p$ Data Preprocessing

3: **function** Updating Global Framework

4:     **commencing weight** $w_0$

5:     **for** $(r_i = 1)$ **to** R **do**

6:         **for** $(c_i = 1)$ **to** C **do**

7:         $w_{r_i+1}^{c_i}$ = Update Local Framework $(c_i, w_{r_i})$

8:         $w_{r_i+1} = \sum_{c_i=1}^{C} w_{D_s}^{c_i} \star w_{r_i+1}^{c_i}$

9: **function** Updating Local Framework $(c_i, w_{r_i})$

10:     **for** $(Epoch = 1)$ **to** $E_p$ **do**

11:         **for** $(B_s \in SizeofBatch)$ **do**

12:         $w = w - \nabla F_i(w)$

13:     **return** $w$

---

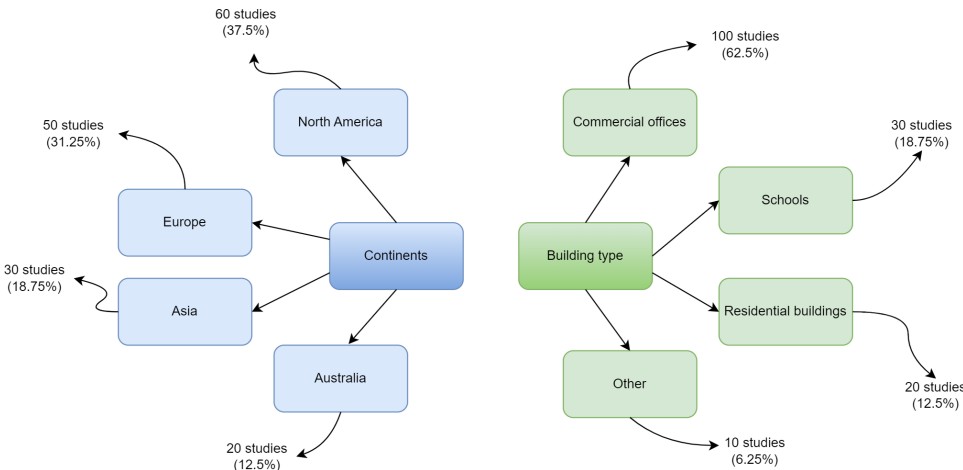

**Figure 2** Geographic distribution of building studies comprising the RP-884 thermal comfort database.

## Experimental dataset

The data is gathered through field surveys covering 160 different building locations worldwide. It is offered as part of the ASHRAE project to establish a thermal comfort preference model. As part of the ASHRAE RP-884 public repository, ASHRAE manages a collection of files from multiple studies conducted by different researchers. Data files are collected from several environmental zones dispersed throughout different geographical areas (*De Dear & Schiller Brager, 2001*). The dataset was chosen because thermal comfort is crucial for human health and productivity. A lack of thermal comfort may cause inconvenience among building occupants. The ASHRAE RP-884 dataset utilized in this study has 12,595 records and 56 characteristics. The creation of an adaptive model is the goal of this dataset. It comprises over 20,000 customer comfort votes from 52 studies conducted across 10 different climatic zones. The dataset contains 55 primary identifiers such as blcode, sub, ash, ACT10, ACT20, ACT30, ACT60, met, clo upholst, insul, and TA_H with three classes: UW, N, and UC. There are 20,757 training record totals. Because the present investigation works with two clients, each with unique data collection, the dataset is split into two windows. Figure 2 presents the overall geographic distribution of building studies comprising the RP-884 thermal comfort database.

## Data preprocessing

Data preprocessing in ML refers to transforming raw information into a format that can be used to develop and improve ML models. Data preprocessing is the first step in ML, preceding the construction of a model. Data preparation is essential to increase data reliability and extract useful information. Real data often lacks specific attributes or patterns, is partial, erroneous (has mistakes or anomalies), and cannot be trusted. Data preparation is crucial in this case because it makes it simpler to arrange, filter, and present raw data in a way that ML models can use. To balance and normalize the data, the dataset is evaluated using two preprocessing techniques: SMOTE and normalization. Using

imitation samples to depict the minority classes, SMOTE is an oversampling technique. The overfitting issue from random oversampling is mitigated partially using this strategy.

*Min-max normalization:* Min-max scaling normalizes the features, retains data integrity, and reduces variation in the uncertain dataset. Models depending on value amplitude need to scale the input attributes. For this reason, the discrete interval of real-valued numerical qualities between 0 and 1 is defined by normalizing. To normalize the data, Eq. (3) is employed.

$$Y_{norm} = \frac{Y_i - Y_{min}}{Y_{max} - Y_{min}}. \tag{3}$$

The dataset is cleaned up and then split into training and testing halves, with 25% of the dataset destined for testing and 75% for training.

## Model architecture

This study looked into DNN's potential for estimating the power consumption of buildings. The DL technique includes the area of learning that uses nonlinear data over several phases through organizational structures, as Dong and Deng mentioned earlier (*Deng & Yu, 2014*). Deep learning is usually used chiefly for pattern recognition and learning. The lower-level elements in the hierarchy of deep architecture are used to build higher-level principles. DL combines NN, pattern identification, and visual design. The DL model forecasts satisfactorily with large data sets. According to *Huang et al. (2013)*, when compared to other ML classifiers, DNNs are extraordinary. The proposed DL model examines the inherent trends in the heating and ventilation load of a structure.

Further, multi-task learning, which considers the attributes of all construction forms using a single-layer deep neural network, makes sense in DL. A DNN is an array of hidden layers in a feed-forward NN. The primary modules in this network are multi-layered self-learning modules. A DNN divides its input and output layers using hidden units. The scalar variable $y_p$ of the next layers can be translated by the hidden unit p using a logistic function to the input $x_p$ beneath. In a DNN network, $i$th neuron $x_i$'s output can be estimated using Eqs. (4) and (5):

$$x_i = f(\xi_i) \tag{4}$$

$$f(\xi_i) = \vartheta + \Sigma_{j\varepsilon\tau_i} - 1 W_i X_j. \tag{5}$$

The $i$th-neuron's potential is denoted by $\xi_i$, whereas the transfer function is represented by $f(\xi_i)$. The following Eq. (6) depicts the transfer function:

$$f(\xi_i) = \frac{1}{1 + exp(-\xi_i)}. \tag{6}$$

The goal cost function where the target values, $y_o$ and $\hat{y}_o$, is determined by the output neurons can be represented as the sum of squared errors as shown in Eq. (7).

$$C = \Sigma 1/2(y_o - \hat{y}_o). \tag{7}$$

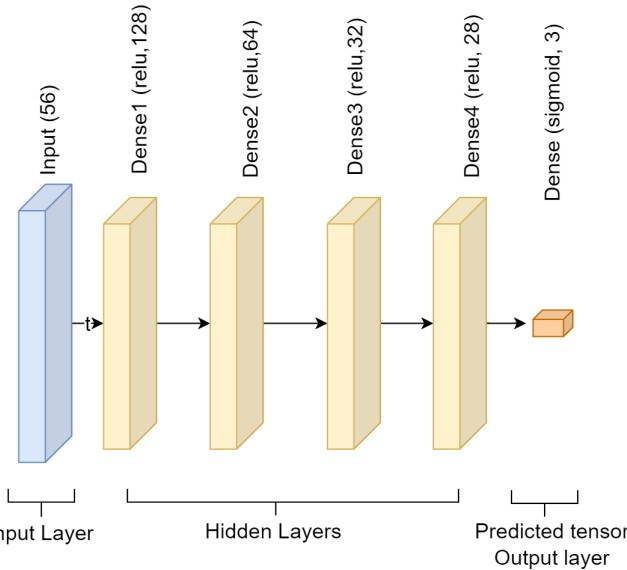

**Figure 3** **Architecture of deep neural network.**

This study used a single-input layer sequential DNN approach. Using 256 units and the relu activation function, the input layer has the shape of a 56. Next, we have the hidden layer, which incorporates four dense layers. Relu is the activation function, and the units that incorporate the four dense layers are 128, 64, 32, and 28. The softmax function is used in the output layer, which comes next and has three units. Each dense layer employs a fully linked layer, and the activation functions softmax and relu to solve the classification issue. The DNN model has employed Adam as an optimizer to compute and minimize loss using sparse_categorical_crossentropy. The model size is 64 kb. Figure 3 presents the DNN model's framework.

## EXPERIMENTAL RESULTS AND ANALYSIS

The following part presents the experiment's results and an assessment of the proposed framework. The consequences of more expansive technique variables are examined in this research. The ASHRAE RP-884 dataset, which has 12,595 data and 56 features, is used for the experiment. In this experiment, there is one server and two clients. The results for every client are verified three times to avoid loss. The client's findings are combined at the server end.

### Server-based training using log data

The FL paradigm uses three primary parameters: one central server and two clients. The FL server directs client selection and collects incoming model improvements during the model training process. Server-logged records are used to train the DNN framework. Logs are confidential and anonymous before the training. The experiment is run three times as though there are three rounds, with various parameters initially for the federated learning model's server setup. Every round of the FL model consists of two phases: the assessment

**Table 1 Proposed framework outcomes of client 1.**

| Rounds | Accuracy % | Precision % | Recall % | F1-Score % |
|---|---|---|---|---|
| Round 1 | 76 | 76 | 85 | 80 |
| Round 2 | 80 | 82 | 87 | 84 |
| Round 3 | 85 | 86 | 90 | 88 |

cycle and the fit round. In the fit round, clients send their training results to the server; both provide their testing results in the evaluation round, and the server aggregates the data. The server evaluates records from N clients and determines that the DNN method has an accuracy of 82.40%.

## Federated training results of client 1

After each round, Table 1 shows an improvement over the previous round. In the first round, 76% accuracy is achieved, which increased to 80% and 85% in rounds 2 and 3, respectively. Similarly, precision is obtained at 76%, 82%, and 86% in rounds 1, 2, and 3, respectively. The recall value is achieved at 85%, 87%, and 90%, and the f1-score is 80%, 84%, and 88%.

Figure 4A showing training and validation accuracy, Fig. 4B loss, and Fig. 4C ROC of round 3. In Fig. 4A, the training and validation accuracy curve is represented. The lowest training accuracy is 0.76% a, reaching a maximum of 0.89% accuracy. The minimum validation accuracy is 0.77%, which increased to 0.85%. Figure 4B graphically represents the training and validation loss graph. The training loss started from 0.55% and decreased to 0.26 and the validation loss started from 0.52% and lessened to 0.40%.

Figure 4C shows the class's receiver operating characteristic (ROC) curve. Three classes were used in the experiment where the ROC curve of class 0 is represented by a blue line with an area of 0.92, the ROC curve of classes 1 and 2 is represented by an orange line with an area of 0.98, and the green line with an area of 0.97 respectively by showing best outcomes on the used dataset for the proposed framework. The ROC curves in the leftmost position indicate a performance improvement. The ROC curves in the upper-left corner show enhanced efficiency.

## Federated training results of client 2

Client 2 performance is less than that of client 1, shown in Table 2, and achieved an accuracy of 72% in the first round, which increased to 80% and 83% after the second and third rounds. Similarly, precision is achieved at 72% in round 1, 80% in round 2, and 84% in round 3. The recall and f1-score are 76% and 75% in round 1, 88% and 84% in round 2, and 89% and 87% in round 3, respectively.

Figure 5 graphically represents the training and validation accuracy curve, loss curve, and ROC curve of round 3 of client 2. Figure 5A indicates the training and validation accuracy curve. The lowest training accuracy is 0.76% following some oscillation between losses and gains, comes 0.90%. The validation accuracy is 0.74% then, after some fluctuation, it increases to 0.83%. The training and validation loss curve is represented in Fig. 5B. the

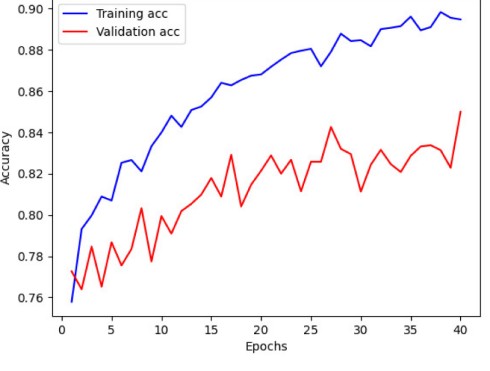
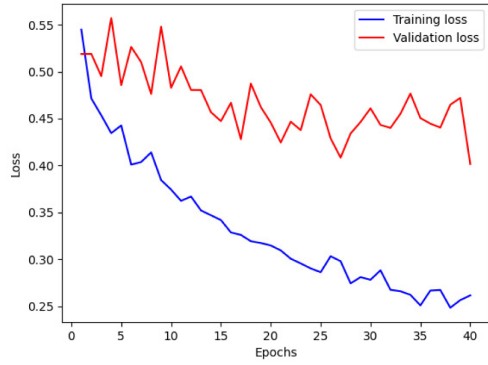

(a) Round 3 validation and training accuracy      (b) Round 3 validation and training loss

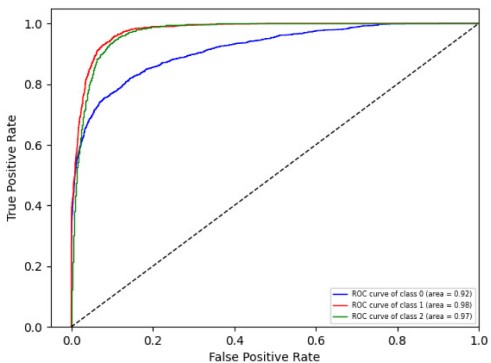

(c) Round 3 score of Receiver Operation Characteristic

**Figure 4**   **Graphical representation of client 1 highest-scoring results.**

**Table 2**   **Proposed model outcomes of client 2.**

| Rounds | Accuracy % | Precision % | Recall % | F1-Score % |
|---|---|---|---|---|
| Round 1 | 72 | 75 | 76 | 75 |
| Round 2 | 80 | 80 | 88 | 84 |
| Round 3 | 83 | 84 | 89 | 87 |

training loss begin from 0.55% and lessen to 0.25. The minimum validation loss is 0.57% and diminishes to 0.45%.

Figure 5C shows the class's ROC curve. Three classes were utilized in the experiment where the ROC curve of class 0 is represented by a blue line with an area of 0.90, the ROC curve of classes 1 and 2 is represented by an orange line with an area of 0.97, and the green line with an area of 0.96 respectively by displaying best results on the used dataset for the proposed approach. The ROC curves in the leftmost position indicate a performance improvement. The ROC curves in the upper-left corner show enhanced efficiency.

The proposed methodology is more successful with fewer false positive and false negative statistics and more persistent, improved real positive and negative results. Last, the dataset

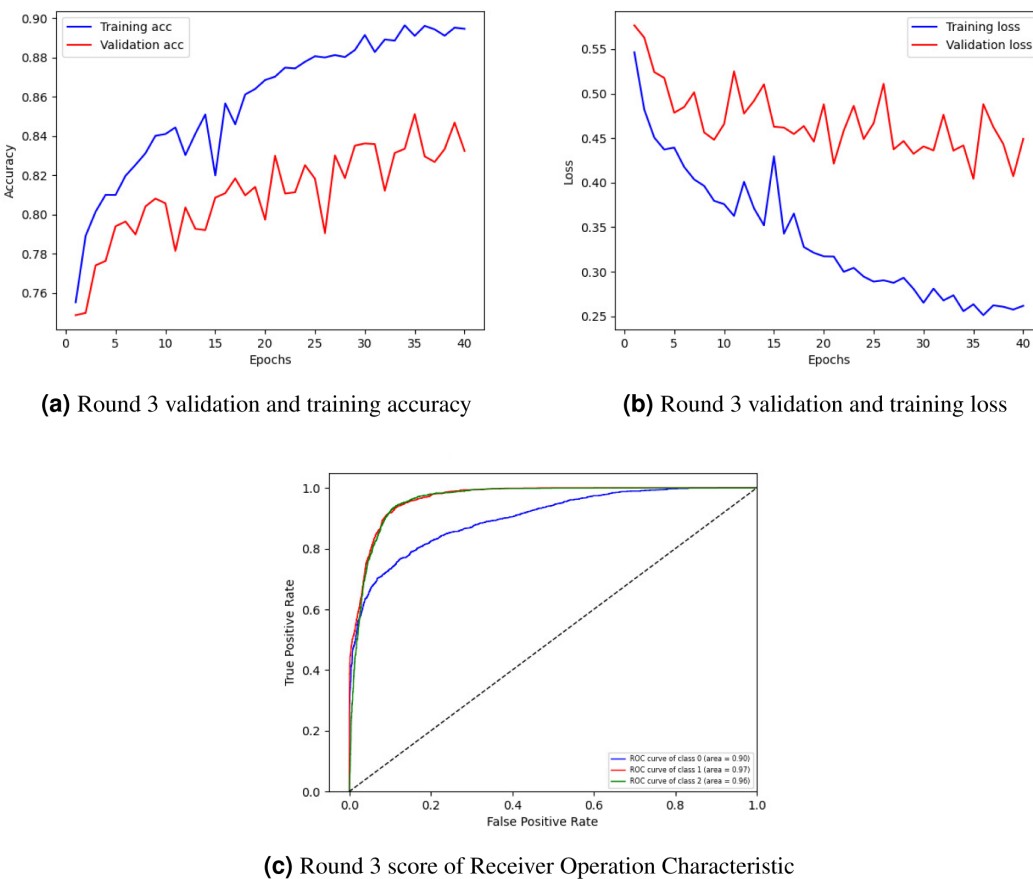

**(a)** Round 3 validation and training accuracy    **(b)** Round 3 validation and training loss

**(c)** Round 3 score of Receiver Operation Characteristic

**Figure 5  Graphical representation of the client's best rating results.**

can be correctly categorized using the current methods. The confusion matrix of the proposed methodology is shown in Fig. 6A, which also provides a performance summary for a classification algorithm. Figure 6B illustrates the proposed methodology's confusion matrix and summarises the classification algorithm's performance so that the dataset can be correctly categorized using the current methods.

## Comparison analysis

The study compares with the existing study (*Zhang & Li, 2023*) as shown in Table 3. According to *Zhang & Li (2023)*, the accuracy is 62.6% accuracy, 57.0% precision, and 59.0% F-score, whereas the proposed method obtained 85.0% accuracy, 86.0% precision, 90.0% recall and 88.0% F-score. The proposed technique performs better in this context than the Base Paper, indicating that our technique outperforms in thermal comfort model prediction for smart buildings.

## Discussion and analysis

An essential component of smart buildings that helps with the development, analysis, and realization of intelligent structures is thermal comfort. These smart buildings' energy

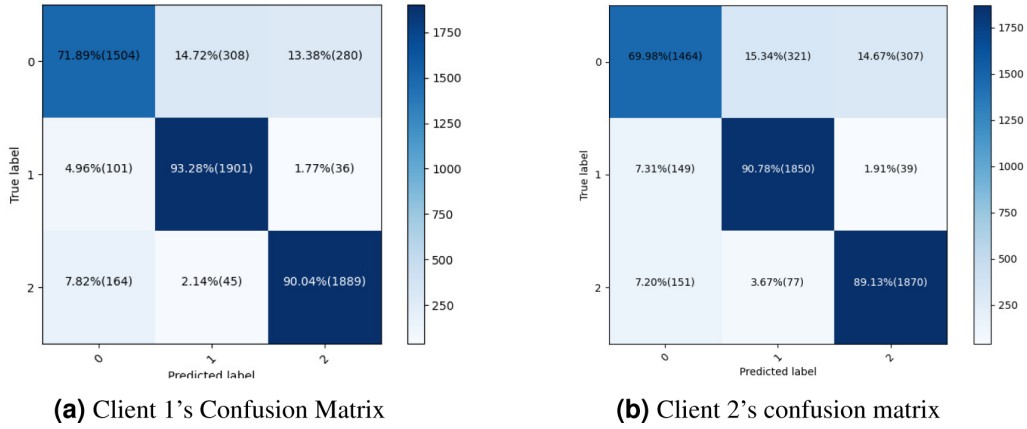

**(a)** Client 1's Confusion Matrix          **(b)** Client 2's confusion matrix

**Figure 6**  Confusion metrics for both clients proposed framework.

**Table 3**  Comparison analysis with existing study.

| Models | Accuracy | Precision | Recall | F1-score |
| --- | --- | --- | --- | --- |
| Transfer learning | 62.6 | 57.0 | NA | 59.0 |
| Proposed model | 85.0 | 86.0 | 90.0 | 88.0 |

usage forecasts are essential since resource efficiency involves complex decision-making procedures. To do the statistical analysis, the accuracy, precision, recall, and F1-measure optimization metrics are used to assess the performance of the proposed model. Results from federated learning models are assessed for performance, significance, and potential for generalization through statistical analysis. Federated learning proposes a distinctive set of challenges compared to traditional centralized ML methods. It raises issues related to delays in communication and bandwidth limits by decentralizing model training among devices. It necessitates an immense amount of resources to transmit model updates from the devices to the central server. It is frequently applied in situations when protecting user privacy is crucial. Complexity arises from ensuring private data stays on local devices while adding to the global model. Securing the model updates and communication channels is essential since decentralized learning can be more susceptible to attacks. In a federated learning environment, devices may differ regarding data distribution, network circumstances, and hardware capacities. This heterogeneity can make the process of upgrading and aggregation more difficult. The central server must compile model updates from various devices. It is difficult to choose an appropriate aggregation approach that combines the inputs from various devices. To overcome all these challenges, we employed the federated deep learning model with various approaches, Including penalty terms to the loss function, and regularisation can lower model complexity. The recommended FDL framework ensures adherence to privacy regulations by implementing responsible and open data processing practices and incorporating privacy-protecting elements. A federated learning-based deep learning model increases prediction performance. This work uses the FDL model to

predict the thermal environment for intelligent constructions. The experiment's outcomes demonstrate that the proposed FDL model outperforms traditional machine learning methods regarding accuracy and efficiency.

## CONCLUSION AND FUTURE SCOPE

Thermal comfort is a critical building automation component that supports analyzing, improving, and implementing intelligent structures. Energy usage is estimated using artificial intelligence approaches, whereas these techniques need a large amount of data to be effective, which can violate privacy. The FDL framework based on a DNN model is proposed in this research to address this issue. The DNN model is trained on two clients, and the server side is combined with 82.40% accuracy. To decrease the over-fitting component, each customer reviewed the findings three times. Client 1 achieved the best result in round three with an accuracy of 85.0%, while client 2 acquired the high result in round three with an accuracy of 83.0%. The recommended approach performs exceptionally well on the tested dataset, as seen by the DNN algorithm's ROC curve average of 98.0%. The outcomes demonstrated that FDL accurately protects the security of client data. The study's future scope will involve expanding its focus to non-ventilated buildings and local thermal comfort, as well as considering other machines and deep learning techniques with multiple datasets to increase the learning efficiency of the HVAC systems and use several statistical tests to ensure the proposed algorithm is reasonable.

### Funding
The authors received no funding for this work.

### Competing Interests
The authors declare there are no competing interests.

### Author Contributions
- Sidra Abbas conceived and designed the experiments, performed the experiments, performed the computation work, prepared figures and/or tables, authored or reviewed drafts of the article, and approved the final draft.
- Shtwai Alsubai conceived and designed the experiments, performed the experiments, performed the computation work, prepared figures and/or tables, and approved the final draft.
- Gabriel Avelino Sampedro analyzed the data, prepared figures and/or tables, authored or reviewed drafts of the article, and approved the final draft.
- Mideth Abisado analyzed the data, prepared figures and/or tables, and approved the final draft.
- Ahmad Almadhor conceived and designed the experiments, performed the experiments, performed the computation work, authored or reviewed drafts of the article, and approved the final draft.

- Tai-hoon Kim analyzed the data, authored or reviewed drafts of the article, and approved the final draft.

## Data Availability

The raw dataset is available in Supplementary Files.

## Supplemental Information

Supplemental information for this article can be found online at http://dx.doi.org/10.7717/peerj-cs.1899#supplemental-information.

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
