# Peer review of "Privacy preserved and decentralized thermal comfort prediction model for smart buildings using federated learning"

_PeerJ Computer Science, doi:10.7717/peerj-cs.1899_

## Round 0.1 · original submission · Major Revisions

Dear Dr. Abbas,

Thank you for your submission to PeerJ Computer Science.

Your article requires a number of Major Revisions.

**Language Note:** The review process has identified that the English language must be improved. PeerJ can provide language editing services - please contact us at copyediting@peerj.com for pricing (be sure to provide your manuscript number and title). Alternatively, you should make your own arrangements to improve the language quality and provide details in your response letter. – PeerJ Staff

Reviewer 1 ·

Basic reporting

In this paper, the author proposes an intelligent building privacy protection and decentralized thermal comfort prediction model based on Federated learning. The model achieves a prediction accuracy of 82.40% while protecting client data security. Overall, this work clearly expresses its goals and contributions. Especially in the introduction and related work, there is a strong correlation between the references and the research purpose of the paper. In the method section, the author provides a detailed introduction to the framework and implementation process of the proposed method. However, the novelty of the paper is poorly. There are still some shortcomings that can be improved in terms of details. For example, in the 1.2 Organization section,?? Is there a formatting issue.

Experimental design

In this section, the author constructed a federated learning scenario consisting of one server and two clients, and conducted experimental verification using the public dataset ASHRAE RP-884. It is commendable that the author provided us with raw data and code. However, the lack of detailed explanations in the paper hinders other researchers from replicating the proposed method. It is recommended to upload the code and data to an open source website and provide a detailed readme file.

Validity of the findings

The author conducted a comparative analysis of the training and validation stages using four evaluation indicators (recall, f1 score, precision, and accuracy). The conclusion statement is sufficient and relevant to the original research question, limited to supporting results.

Additional comments

no comment

Reviewer 2 ·

Basic reporting

The paper has several typos. Authors need to proofread the paper to eliminate all of them.

The introduction should clearly explain the key limitations of prior work that are relevant to this paper.

Contributions should be highlighted more. It should be made clear what is novel and how it addresses the limitations of prior work.

The authors should explain clearly what the differences are between the prior work and the solution presented in this paper.

Experimental design

it is important to better explain the design decisions (e.g. why the solution is designed like that)

It is necessary to discuss the complexity of the proposed solution. For federated learning, one of the common drawbacks is the communcation overhead. authors should explain how they can solve this issue

Moreover, model size is another important issue, thus, authors should give precise information about local hardware and model size, training size, communication overhead etc.

The authors should explain how tthey selected their model configuration.

The experiments should be updated to include some comparison with newer studies.

The experiments have been carried with only one dataset. It is necessary to add more datasets so as to make experiments more convincing.

Some additional experiments are required:
- Scalability
- Runtime

Validity of the findings

There is not enough discussion of the experimental results.

A statistical analysis should be carried out to demonstrate that the experimental results are significant.

authors should discuss the future work or research opportunities.

Additional comments

Their proposed work discussion is weak

The novelty is not guaranteed.

Their work is not compared with state-of-the-art approaches nor related studies.

Their experiments leak from the descriptive and statistical analysis.

Reviewer 3 ·

Basic reporting

The paper is an interesting approach in proposing a Federated Deep Learning (FDL) framework based on a Deep Neural Network (DNN) model. the study employs the ASHRAE RP-884 standard dataset, which is available to the general public. The data is normalized using the min-max normalization technique, and Synthetic Minority Over-Sampling Technique (SMOTE) is used to enhance the minority class’s interpretation.

Experimental design

In a satisfactory manner, the basic purpose of the research has been described, but with only some crucial comments that should be taken into consideration:
• The federated deep learning paradigm proposed by the authors consists of only two clients. Why the authors choose two clients not three, four, or five. Does the increase of the number of clients will increase the performance of the FDL?

• The author used the ASHRAE RP-884 standard dataset. It is better to add more details about the dataset with a screenshot from it.

Validity of the findings

Any experimental results are presented to highlight and validate the proposed approach with a comparison among previous work.
the authors do not represent how this results is different from any previous studies as a comparison. How to validate the results?

Additional comments

Only one comment in the list of references, all the references seem to be good but there is NO references from 2023. We advise the author to add at least FOUR recent references belonging to 2023 to enrich the manuscript.

---

## Round 0.2 · accepted · Accept

Your manuscript has been accepted for publication in PeerJ Computer Science.

Reviewer 1 ·

Basic reporting

In this paper, the author proposes an intelligent building privacy protection and decentralized thermal comfort prediction model based on Federated learning. The model achieves a prediction accuracy of 82.40% while protecting client data security. Overall, this work clearly expresses its goals and contributions, and has a certain degree of novelty. The authors have addressed the reviewers' comments and concerns in their revised manuscript. The structure conforms to PeerJ standards.

Experimental design

Methods described with sufficient detail & information to replicate. The author has provided raw data and experimental code, as well as detailed readme files, which have met the standard requirements of the journal.

Validity of the findings

Conclusions are well stated, linked to original research question & limited to supporting results. The author verified the feasibility of the proposed solution through security analysis and performance evaluation.

Additional comments

No.

Reviewer 2 ·

Basic reporting

The authors have diligently executed all the requested modifications

Experimental design

The authors have diligently executed all the requested modifications

Validity of the findings

The authors have diligently executed all the requested modifications

Additional comments

The authors have diligently executed all the requested modifications